# Normality-Guided Distributional Reinforcement Learning for Continuous Control

**Ju-Seung Byun**[*]                                                                    *byun.83@osu.edu*
*Department of Computer Science and Engineering*
*The Ohio State University*

**Andrew Perrault**                                                                    *perrault.17@osu.edu*
*Department of Computer Science and Engineering*
*The Ohio State University*

**Reviewed on OpenReview:** *https://openreview.net/forum?id=z27hb0rmLT*

## Abstract

Learning a predictive model of the mean return, or value function, plays a critical role in many reinforcement learning algorithms. Distributional reinforcement learning (DRL) has been shown to improve performance by modeling the value *distribution*, not just the mean. We study the value distribution in several continuous control tasks and find that the learned value distribution is empirically quite close to normal. We design a method that exploits this property, employing variances predicted from a variance network, along with returns, to analytically compute target quantile bars representing a normal for our distributional value function. In addition, we propose a policy update strategy based on the correctness as measured by structural characteristics of the value distribution not present in the standard value function. The approach we outline is compatible with many DRL structures. We use two representative on-policy algorithms, PPO and TRPO, as testbeds. Our method yields statistically significant improvements in 10 out of 16 continuous task settings, while utilizing a reduced number of weights and achieving faster training time compared to an ensemble-based method for quantifying value distribution uncertainty.

## 1 Introduction

In reinforcement learning, an agent receives rewards by interacting with the environment and updates their policy to maximize the cumulative reward or return. The return often has a high variance, which may make training unstable. To reduce the variance, actor-critic algorithms (Thomas, 2014; Schulman et al., 2015b; Mnih et al., 2016; Gu et al., 2016) use a learned value function, a model of the return that serves a baseline to reduce variance and speed up training.

The standard approach to learning a value function estimates a single scalar value for each state. Several recent studies learn a *value distribution* instead, attempting to capture the randomness in the interaction between the agent and environment. One method for achieving this involves using multiple value functions (referred to as an ensemble) so that these functions collectively represent the value distribution (Lee et al., 2020). An alternative method requires the value function to produce multiple outputs, represented as quantile bars (Bellemare et al., 2017a; Barth-Maron et al., 2018; Dabney et al., 2017; Singh et al., 2020; Yue et al., 2020). These techniques, known as distributional reinforcement learning (DRL), have proven to enhance stability and expedite the learning process. The target for generating quantile bars is commonly derived from the distributional Bellman optimality operator, and the distributional value function is fitted to the target with the quantile regression (Bellemare et al., 2017a; Dabney et al., 2017).

---

[*]Corresponding author

Our goal is to derive quantile bars based on the Markov chain central limit theorem (MC-CLT) rather than the distributional Bellman optimality operator, and to introduce a novel uncertainty measure absent in standard distributional value functions, applying it to policy updates. The MC-CLT indicates that in continuous tasks, the variance of the return keeps decreasing as the timestep goes. Moreover, when sufficient remaining timesteps exist, the return conforms to a normal distribution. We examine the value distribution trained using quantile regression and find, in continuous tasks, the predictions of the distribution closely resemble normal distributions, as described by the MC-CLT and depicted in Figure 1. This contrasts with the distributional Bellman optimality operator, which causes the variance of the value distribution to increase over timesteps (Figure 2), due to the multiplication of $\gamma < 1$ with the next state and action Q value (Equation 4). Section 4.2 provides a detailed example of this variance increase.

In this paper, we assume that the value distribution of continuous tasks is governed by the MC-CLT rather than the distributional Bellman optimality operator. To appropriately guide the distributional value function, we provide analytically computed quantile targets representing normal distributions, grounded in variances sourced from a variance network (Nix & Weigend, 1994; Kendall & Gal, 2017) and returns. We assess the accuracy of the current predictions made by the distributional value function. We aim to leverage distributional information, prioritizing values that are accurately estimated during policy updates.

As we consider the reliable values for policy updates, we incorporate existing actor-critic algorithms (Lee et al., 2020; Mai et al., 2022) that factor in uncertainty during policy updates as a comparison. These algorithms typically minimize the influence of samples with high variances. However, some states inherently possess elevated variances. For instance, the true variances are higher near the initial timestep because the expected return from each timestep gets smaller as timesteps goes on (Appendix D). The question arises: *Should states having higher variances have less impact on policy updates?* Although the predicted variance is *correct* for a state, its impact could be lessened simply because its variance is high. In contrast, we determine the impact of policy updates by evaluating how closely our distributional value function predicts the quantiles for normals. Given that our distributional value function is fitted to a normal quantile target, a well-trained distributional value function should yield quantiles that closely align with the shape of the normal distribution. This alignment indicates the function's accuracy in reflecting its models' underlying distribution. In Appendix E, we provide additional discussion on tasks where the normality assumption may not be applicable, as well as experiments in the opposite direction, where samples that are not closely aligned are given higher weights for exploration.

In summary, our key contributions are:

- We examine that the distributional value function approximates a normal for continuous tasks, and this normal's variance increases with each timestep. To address this, we guide the distributional value function to produce quantiles representing a normal distribution, leveraging variances sourced from the variance network (Nix & Weigend, 1994; Kendall & Gal, 2017) and sampled returns.

- Under the normality assumption, we measure the uncertainty (or correctness) of the value distribution for each state, rather than relying the uncertainty derived from multiple value functions (ensemble). We propose an uncertainty-based policy update strategy that assigns a high weight to a correctly predicted state.

- We provide an empirical validation that uses PPO (Schulman et al., 2017) and TRPO (Schulman et al., 2015a) on several continuous control tasks. We compare our methods to standard algorithms (PPO and TRPO) as well as the ensemble-based approach. We find that our method exhibits better performance than the ensemble-based method in 10/16 tested environments, while using half as many weights and training twice as fast.

## 2   Related Work

We briefly review related work on distributional reinforcement learning (DRL) and supporting methods. Bellemare et al. (2017a) studied the distribution of the value network for deep Q-learning and proposed C51, a DRL algorithm that uses a categorical distribution to model the value distribution. C51 produced

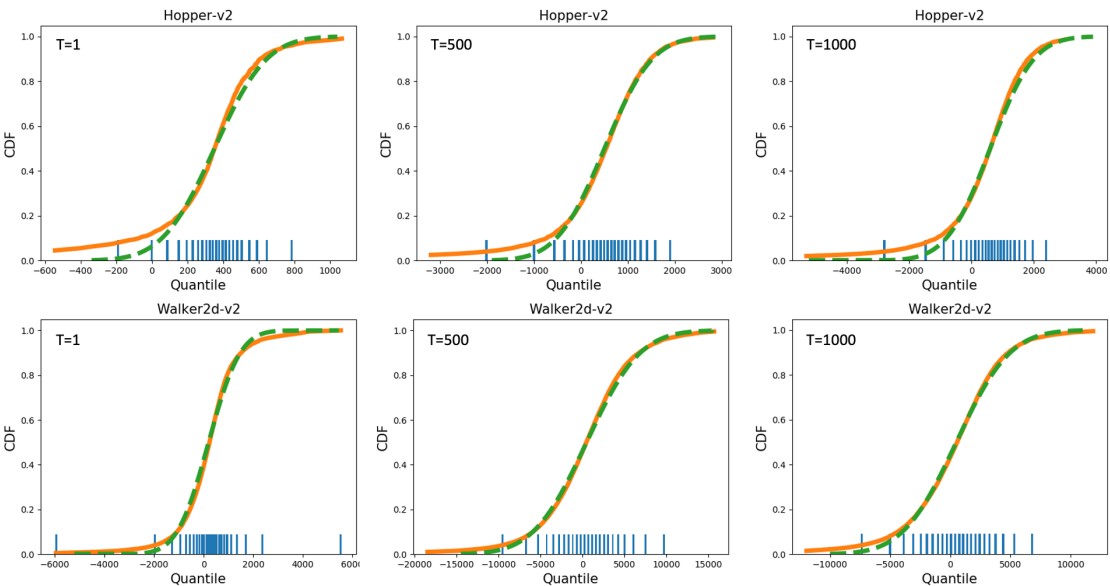

Figure 1: Results of trained the distributional value function $V^{D,\pi}$ with the Huber quantile regression. Each line represents different timesteps: the initial step $T = 0$, the intermediate step $T = 500$, and the last step $T = 1000$. Within each cell, the quantile output of $V^{D,\pi}$ is represented by bars, the corresponding cumulative distribution function (CDF) is depicted in orange, and the computed normal CDF based on the bars is shown in green. The policies are updated with PPO and $V^{D,\pi}$s are updated with the quantile Huber loss. The number of quantile bars is 200, but only 45 are shown for visualization. Notably, for continuous tasks, the distribution predicted by $V^{D,\pi}$ aligns remarkably well with the normal distribution, even at the final step.

state-of-the-art performance on the Arcade Learning Environment (Bellemare et al., 2013), but required the parameters of the modeling distribution to be fixed in advance. Dabney et al. (2017) proposed QR-DQN, which leverages quantile regression and allows for an adaptive distribution structure. In addition, they resolved a key theoretical question with KL divergence ($D_{KL}$), which could lead to non-convergence, by replacing it with Wasserstein distance metric and showing convergence of the distributional value network. While these methods focus solely on the mean of the value distribution, our approach seeks to leverage the distributional characteristics through the normality assumption as well.

Several extensions of DRL to continuous tasks have been proposed, including distributed distributional deterministic policy gradients (D4PG) (Barth-Maron et al., 2018), sample-based distributional policy gradient (SDPG) (Singh et al., 2020) and implicit distributional actor-critic (IDAC) (Yue et al., 2020). Our method is model agnostic and compatible with any existing DRL algorithm that learns the value distribution with a single neural network.

Compared to supervised learning, reinforcement learning has an unstable learning process, and uncertainty is often modeled to make RL learning more stable. We can decompose uncertainty into aleatoric and epistemic (Hora, 1996; Kiureghian & Ditlevsen, 2009). Aleatoric comes from the stochasticity of data and is often measured with the attenuation loss function (Nix & Weigend, 1994). Epistemic uncertainty is caused by the limits of generalization of models and can be measured by dropout (Kendall & Gal, 2017) or ensemble methods (Lakshminarayanan et al., 2016). The RL community models these two types of uncertainty to update the model safely or to support sparse reward functions (Bellemare et al., 2016; Strehl & Littman, 2005; Burda et al., 2018). Clements et al. (2019) penalizes actions with high aleatoric uncertainty, and epistemic uncertainty is used to guide exploration. To our knowledge, Moerland et al. (2017) is the only previous work assuming a normal distribution for the return distribution, which they use to construct a bootstrap estimate of combined uncertainty. SUNRISE (Lee et al., 2020) measures epistemic uncertainty with an ensemble technique and uses it to reduce the gradient of a sample with high uncertainty. Mai

et al. (2022) leverage the two types of uncertainty with the combination of ensemble and attenuation loss. We utilize quantile bars, which represent the return's distribution, as a proxy for epistemic uncertainty. This proxy is less resource-intensive in terms of training compared to ensemble methods, which significantly increase training costs (see Appendix of Mai et al. (2022)).

## 3 Preliminaries

### 3.1 Reinforcement Learning

We consider an infinite-horizon Markov decision process (MDP) (Puterman, 2014; Sutton & Barto, 2018) defined by the tuple $\mathcal{M} = (\mathcal{S}, \mathcal{A}, \mathcal{P}, R, \gamma, \mu)$. The agent interacts with an environment and takes an action $a_t \in \mathcal{A}$ according to a policy $\pi_\theta(a_t|s_t)$ parameterized by $\theta$ for each state $s_t \in \mathcal{S}$ at time $t$. Then, the environment changes the current state $s_t$ to the next state $s_{t+1}$ based on the transition probability $\mathcal{P}(s_{t+1}|s_t, a_t)$. The reward function $R : \mathcal{S} \times \mathcal{A} \to \mathbb{R}$ provides the reward for $(s_t, a_t)$. $\gamma$ denotes the discount factor and $\mu$ is the initial state distribution for $s_0$. The goal of a reinforcement learning (RL) algorithm is to find a policy $\pi_\theta$ that maximizes the expected cumulative reward, or return:

$$\theta^* = \operatorname*{argmax}_\theta \mathbb{E}_{\substack{s_0 \sim \mu \\ a_t \sim \pi_\theta(\cdot|s_t) \\ s_{t+1} \sim \mathcal{P}(\cdot|s_t, a_t)}} \left[ \sum_{t=0}^\infty \gamma^t R(s_t, a_t) \right]. \tag{1}$$

In many RL algorithms, a value function $V^\pi(s_t)$ is trained to estimate the expected return under the current policy, $\mathbb{E}[\sum_{t'=t}^\infty \gamma^{t'} R(s_{t'}, a_{t'})]$, and is updated with the target:

$$V^\pi(s_t) = R(s_t, a_t) + \gamma V^\pi(s_{t+1})$$
$$\text{where } s_{t+1} \sim \mathcal{P}(\cdot|s_t, a_t) \tag{2}$$

### 3.2 Distributional Reinforcement Learning

In distributional reinforcement learning (DRL), Bellemare et al. (2017b) introduced an action-state distributional value function $Q^{D,\pi}(s, a)$ which models the distribution of returns for an agent taking action $a$ and then following policy $\pi$. We model the distribution by having $Q^{D,\pi}$ output $N$ quantile bars $\{q_0, q_1, ..., q_{N-1}\}$. Similar to the Bellman optimality operator $\mathcal{T}^\pi$ (Watkins & Dayan, 1992) (Equation 3) for the state-action value function $Q^\pi$, $Q^{D,\pi}$ is updated by a distributional Bellman optimality operator (Equation 4):

$$\mathcal{T}^\pi Q^\pi(s_t, a_t) = \mathbb{E}[R(s_t, a_t)] + \gamma \mathbb{E}_{s_{t+1} \sim \mathcal{P}(\cdot|s_t, a_t)}[\max_{a_{t+1}} Q^\pi(s_{t+1}, a_{t+1})]$$
$$\text{where } s_{t+1} \sim \mathcal{P}(\cdot|s_t, a_t), a_{t+1} = \operatorname{argmax}_{a_{t+1}} Q^\pi(s_{t+1}, a_{t+1}) \tag{3}$$

$$\mathcal{T}^\pi Q^{D,\pi}(s_t, a_t) = R(s_t, a_t) + \gamma Q^{D,\pi}(s_{t+1}, a_{t+1})$$
$$\text{where } s_{t+1} \sim \mathcal{P}(\cdot|s_t, a_t), a_{t+1} = \operatorname{argmax}_{a_{t+1}} \mathbb{E}[Q^{D,\pi}(s_{t+1}, a_{t+1})] \tag{4}$$

### 3.3 Markov Chain Central Limit Theorem

We introduce the Markov Chain Central Limit Theorem (MC-CLT) (Jones, 2004) (Theorem 1). We leverage this theorem to argue that returns are normally distributed in some continuous environments.

**Theorem 1** (Markov Chain Central Limit Theorem). *Let $X$ be a Harris ergodic Markov chain on $X$ with stationary distribution $\rho$ and let $f : X \to \mathbb{R}$ is a Borel function. If $X$ is uniformly ergodic and $\mathbb{E}_\rho[f^2(x)] < \infty$*

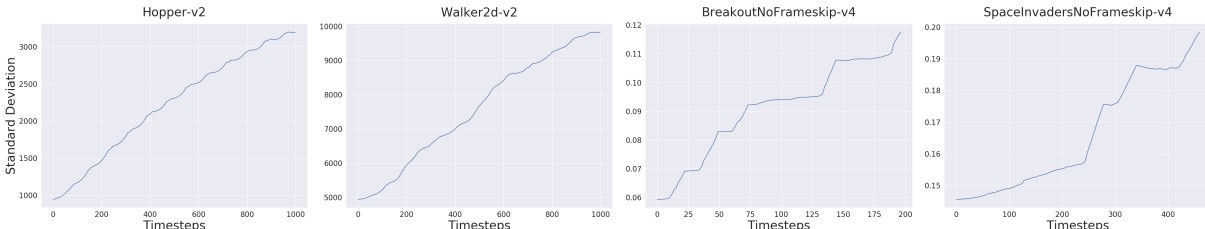

Figure 2: The standard deviation computed from each of the distributional value functions trained on the distributional Bellman optimality operator (Equation 4) with exponential smoothing. The estimated standard deviation increases as timestep increases.

*then for any initial distribution, as $n \to \infty$,*

$$\sqrt{n}(\bar{f}_n - \mathbb{E}_\rho[f]) \xrightarrow{d} \mathcal{N}(0, \sigma^2),$$

$$where \quad \bar{f}_n = \frac{1}{n}\sum_{i=1}^{n} f(X_i) \xrightarrow{a.s} \mathbb{E}_\rho[f]$$

$$\sigma^2 = \mathrm{Var}_\rho(f(X_1)) + 2\sum_{k=1}^{\infty} \mathrm{Cov}_\rho(f(X_1), f(X_{1+k})). \tag{5}$$

To relate this to RL, we let $f$ be the reward function $R$ and $X$ be the state-action space. If the Markov chain induced by a policy is Harris ergodic, we expect that the theorem holds (as rewards being a Borel function is a mild condition). In detail, a return $G(s_t)$ is consists of the sum of the rewards $\sum_{i=t}^{T} R(s_t, a_t)$. Thus, for sufficiently large $T$, the distribution of $G(s_t)$ tends towards a normal distribution. This might not consistently be the case, particularly when $t$ closely approaches $T$ and $R(s_t, \cdot)$ deviates from a normal for each possible action. Nevertheless, a distributional value function, trained using the Huber quantile regression, outputs a distribution that closely represents a normal for the final timestep, even for *terminal* states (Figure 1). It's also worth noting that the presence of transient states can disrupt Harris ergodicity. However, this issue is not prevalent in many continuous control tasks, such as robotics environments in OpenAI Gym, since the optimal policy is stable and repetitive.

## 4 Approach

In this paper, we introduce a method that leverages the distributional property of value distribution, assuming normality in continuous tasks. The objective of this approach is to determine the impact of individual samples on policy updates. We argue that existing quantile distributional value functions yield estimates where variance *increases* with timestep. To handle this issue, we guide our distributional value function $V^{D,\pi}$ using quantile bars of a normal distribution derived from variance and return. Furthermore, we discuss how to measure the uncertainty (or accuracy) of the current estimation of $V^{D,\pi}$ for each state and incorporate this measure into policy updates. We also further discuss scenarios where a task doesn't satisfy the normality condition in Appendix E.

### 4.1 State Distributional Value Function

We first similarly introduce a state distributional value function $V^{D,\pi}(s)$ that also outputs quantile bars for state $s$. The distributional value function $V^{D,\pi}(s)$ for the general actor-critic policy gradient is only dependent on the state. To be specific, the target for $V^{D,\pi}(s_t)$ is $R(s_t, a_t) + V^{D,\pi}(s_{t+1})$ similar to Equation (4). The quantile bars of $V^{D,\pi}(s_t)$ is shifted by the amount of $R(s_t, a_t)$. This let us use the actor-critic algorithms such as Policy Optimization (PPO) (Schulman et al., 2017) and Trust Region Policy Optimization (TRPO) (Schulman et al., 2015a).

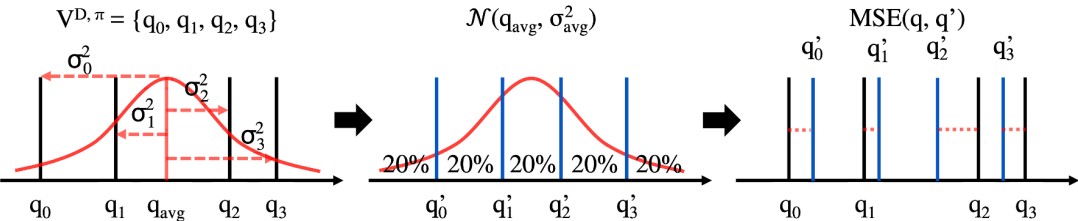

Figure 3: Illustration of how we measure uncertainty. When $N = 4$, there are 4 quantile bars $\{q_0, q_1, q_2, q_3\}$ as an output of $V^{D,\pi}$. Compute the mean $q_{avg}$ of the quantile bars, and find $\sigma_i$ such that $P(Z \leq \frac{q_i - q_{avg}}{\sigma_i}) = 0.2 *$ $(i+1) = \frac{i+1}{N+1}$ for $i \in \{0, 1, 2, 3\}$. We consider $V^{D,\pi}$ as approximately $\mathcal{N}(q_{avg}, \sigma_{avg}^2)$ where $\sigma_{avg} = \frac{1}{4} \sum_{i=0}^{i=3} \sigma_i$. We find the exact quantile location $\{q_0', q_1', q_2', q_3'\}$ of $\mathcal{N}(q_{avg}, \sigma_{avg}^2)$, then measure how much two types of quantiles are different with mean squared error.

## 4.2 Analysis of Distributional Value Function

Motivated by Theorem 1, we examine the learned distributional value functions $V^{D,\pi}$ across different states and at several timesteps at convergence (Figure 1). These distributions appear close to normal distributions.

Furthermore, our observation reveals that the learned $V^{D,\pi}$ and $Q^{D,\pi}$ exhibit an increase in variances as timesteps progress, contradicting Theorem 1. The distributional Bellman optimality operator (Bellemare et al., 2017b) causes the variance of the distributional value function to increase with timestep as the discount factor shrinks quantiles. To illustrate, suppose $Q^{D,\pi}(s_{t+1}, a_{t+1}) = \{q_0, q_1, ..., q_{N-1}\}$ at timestep $t + 1$. Then, the target for $Q^{D,\pi}(s_t, a_t)$ under the distributional Bellman operator will be $\{R(s_t, a_t) + \gamma q_0, R(s_t, a_t) + \gamma q_1, ..., R(s_t, a_t) + \gamma q_{N-1}\}$, i.e., $\{q_0, q_1, ..., q_{N-1}\}$ is shifted by $R(s_t, a_t)$ and shrunk by $\gamma < 1$. Consequently the range of the distribution for timestep $t$ is narrower than the range for timestep $t + 1$.

We test this empirically by training distributional value functions with PPO and QR-DQN using the distributional Bellman optimality operator (Equation 4) and graphing the estimated standard deviation from quantile output for each state (Figure 2). Indeed, the variance estimates appear to increase with timestep on average. We now propose a method that utilizes the variance network and returns to generate quantiles for the distributional value function.

## 4.3 Variance of Normal Distribution

Variance network ($\sigma_\psi^2$), introduced by Nix & Weigend (1994) and named by Mai et al. (2022), are designed to capture both the mean and Gaussian error (variance). In line with the normal assumption, the variance network allows us to estimate the variance of a Gaussian distribution. We leverage a variance network to estimate a variance for each state when computing quantile bars.

In the original variance network, the mean and variance are predicted jointly within a single network. We separate these predictions: the variance network $\sigma_\psi^2$ solely predicts the variance, while the mean is predicted by $V_\theta^{D,\pi}$ like the original actor critic method.

$$\mathcal{L}_{\sigma^2}(\psi) = \frac{1}{T} \sum_{i=1}^{T} \frac{1}{2\sigma_\psi^2(s_i)} ||q_{avg}(s_i) - G(s_i)||^2 + \frac{1}{2} \ln \sigma_\psi^2(s_i) \tag{6}$$

where $q_{avg}(s)$ is the mean of $V_\theta^{D,\pi}(s)$ and $G(s)$ is the sum of sampled rewards from the state $s$ (return). This separation of prediction eliminates the need for additional hyperparameter tuning that would have been required for the simultaneous prediction of mean and variance, as observed in Mai et al. (2022). Although the numerator of Equation 6 is separated from $\psi$, the update is performed in the direction of increasing $\sigma_\psi^2$ when the error of the numerator is significant. Conversely, if the error is sufficiently small, $\sigma_\psi^2$ is decreased. $\sigma_\psi^2$ is trained to predict variance in a manner consistent with the original variance network.

### 4.4 Target for Distributional Value Function

We provide the quantile bars of an approximated normal distribution to $V_\theta^{D,\pi}$ with the variance from $\sigma_\psi^2$ above and returns. To compute the target quantile bars, we need the mean of the approximated normal distribution. The mean can be TD target or the sum of rewards $G(s)$ from $s$ to end. Here, we use return $G$ for the sake of notation simplicity. To obtain a quantile target for $V^{D,\pi}$, we first precompute the $Z$ values of the standard normal distribution. For example, when $N = 4$, $Z = \{-0.841, -0.253, 0.253, 0.841\}$, i.e., $P(X <= Z[i]) \approx 0.2 * (i+1) = \frac{i+1}{N+1}$ for $i \in \{0, 1, 2, 3\}$. Note that we use $N = 8$ quantile bars for all experiments in practice, as we empirically find that $N \geq 8$ provides satisfactory performance. Then, we compute the target quantile bars $q_t'$ of $V_\theta^{D,\pi}(s_t)$ such that $q_t' = \{R_t + \sigma_\psi Z[i] \,|\, i \in \{0, 1, ..., N-1\}\}$ and fit $V_\theta^{D,\pi}$ to $q_t'$ by quantile Huber loss (Aravkin et al., 2014). The quantile Huber loss makes $V_\theta^{D,\pi}$ output quantile bars that match the definition of quantiles. The Huber loss (Huber, 1964) is as follows:

$$\mathcal{L}_\kappa(u) = \begin{cases} \frac{1}{2}u^2, & \text{if } |u| < \kappa \\ \kappa(|u| - \frac{1}{2}\kappa), & \text{otherwise.} \end{cases} \tag{7}$$

The quantile Huber loss is an asymmetric variant of the Huber loss.

$$\mathcal{L}_q(\theta) = \frac{1}{TN} \sum_{t=0}^{T} \sum_{i=0}^{N-1} \rho_{\tau_i}^\kappa (q_i(s_t) - q_{t,i}'),$$

$$\text{where } \rho_{\tau_i}^\kappa(u) = |\tau_i - \delta_{\{u<0\}}| \mathcal{L}_\kappa(u) \tag{8}$$

$$\tau_i = \frac{i+1}{N+1} \text{ for } i = \{0, 1, ..., N-1\}$$

### 4.5 Uncertainty Weight

Since $V^{D,\pi}$ is a distributional value function, we can use the properties of the distribution to calculate the uncertainty weight for policy updates. We measure the distance between the current quantile output of $V^{D,\pi}$ and the normal distribution computed based on $V^{D,\pi}$'s quantile output. We evaluate how close $V^{D,\pi}$ is to the target normal distribution to judge whether $V^{D,\pi}$ generalizes well for the visited states, under the hypothesis that a more accurate $V^{D,\pi}$ will lead to better policy improvement. For example, when $V^{D,\pi}$ does not align with normal quantile bars for a state, the discrepancy between $V^{D,\pi}$ and its target becomes substantial. We discount the impact of such samples having high discrepancy (or high uncertainty) on policy updates to focus more on reliable values that are accurately estimated from the distributional value function.

Figure 3 illustrates this when the number of quantile bars is $N = 4$ for simplicity. In practice, we use $N = 8$ quantile bars for all experiments, as we empirically find that $N \geq 8$ provides similar satisfactory performance. If the prediction of $V^{D,\pi}$ is reliable, it should be symmetrical with respect to the mean of the quantile bars from our normality assumption, and the location of the quantile bars should follow a normal distribution. We first find the mean $q_{\text{avg}}(s) = \frac{1}{N} \sum_{i=0}^{i=N-1} q_i(s)$ of the quantile bars and calculate the standard deviations $\{\sigma_0, \sigma_1, ..., \sigma_{N-1}\}$ for each quantile such that $P(Z \leq \frac{q_i - q_{\text{avg}}}{\sigma_i}) = \frac{i+1}{N+1}$ for $i \in \{0, 1, ..., N-1\}$. We consider $V^{D,\pi}$ is trying to approximate $\mathcal{N}(q_{\text{avg}}(s), \sigma_{\text{avg}}^2)$ where $\sigma_{\text{avg}} = \frac{1}{N} \sum_{i=0}^{i=N-1} \sigma_i$.

To measure the uncertainty $w$ for policy updates, we find the locations of each quantile $\{q_0'(s), ..., q_{N-1}'(s)\}$ based on $\mathcal{N}(q_{\text{avg}}(s), \sigma_{\text{avg}}^2)$. The better $V^{D,\pi}$ resembles the normal distribution, the smaller the difference between these two types of quantile bars. Therefore, we use the mean sqaured error $E = \sum_{i=0}^{i=N-1} (q_i(s) - q_i'(s))^2$, and we compute the uncertainty weight $w(s)$ as follows:

$$w(s) = \sigma(-E * T) + 0.5 \tag{9}$$

where $T > 0$ is a temperature and $\sigma$ is the sigmoid function. Although Lee et al. (2020) suggest values for $T$, it seems that finding the reasonable value for $T$ still requires some tuning. We instead set a target uncertainty weight and perform a parametric search by adjusting $T$ to achieve that target weight (Appendix B).

---

**Algorithm 1** MC-CLT with Uncertainty Weight

---

1: **Input:** policy $\pi$, distributional value function $V_\theta^{D,\pi}$, variance network $\sigma_\psi^2$, rollout buffer $B$
2: Initialize $\pi$, $V^{D,\pi}$, and $\sigma_\psi^2$
3: **for** $i = 1$ to epoch_num **do**
4:      **for** $j = 1$ to rollout_num **do**
5:          $a_t \sim \pi(\cdot|s_t)$
6:          $s_{t+1} \sim \mathcal{P}(\cdot|s_t, a_t)$
7:          Compute $\mathcal{N}(q_{\text{avg}}(s), \sigma_{\text{avg}}^2)$ with $V^{D,\pi}(s_t) = \{q_0(s_t), q_1(s_t), ..., q_{N-1}(s_t)\}$
8:          Find $q'(s_t)$ from $\mathcal{N}(q_{\text{avg}}(s), \sigma_{\text{avg}}^2)$
9:          Compute the mean squared error $E = \sum_{i=0}^{N-1}(q_i(s_t) - q_i'(s_t))^2$
10:          Store $(s_t, a_t, r(s_t, a_t), E, V^{D,\pi}(s_t), \sigma_\psi^2(s_t))$ in $B$
11:          **if** $s_{t+1}$ is terminal **then**
12:             Reset env
13:          **end if**
14:      **end for**
15:      Perform the parametric search to find the temperature $T$ (Appendix B)
16:      Compute target quantiles for $V^{D,\pi}$ with the stored return and $\sigma_\psi^2$
17:      Minimize $\mathcal{L}_{\sigma^2}(\psi)$ and $\mathcal{L}_q(\theta)$
18:      Optimize $\pi$ with a policy objective scaled by $w$
19: **end for**

---

### 4.6 Policy Update with Uncertainty Weight

We leverage the uncertainty weights $w$ to update the policy to mainly focus on values for which the current $V^{D,\pi}$ predicts correctly. Each reinforcement learning algorithm has a policy objective function. We scale the objective function with the uncertainty weight to adjust the influence of the gradient for each sample. For TRPO (Schulman et al., 2015a),

$$\underset{\theta}{\text{maximize}} \ \mathbb{E}_t\left[w(s_t)\frac{\pi_\theta(a_t|s_t)}{\pi_{\theta_{old}}(a_t|s_t)}\hat{A}_t\right]$$
$$\text{subject to} \ \mathbb{E}_t\left[D_{KL}\left(\pi_{\theta_{old}}(\cdot|s_t)||\pi_\theta(\cdot|s_t)\right)\right] \leq \delta, \tag{10}$$

where $\pi_\theta$ is a policy and $\hat{A}_t$ is an advantage estimator at timestep $t$. The more accurate the prediction of $V^{D,\pi}$, the closer the value of $w$ is to 1; hence the gradient is more significant than other inaccurate samples. Note that we normalize the advantages when we sample a batch for optimization in practice. Therefore, the magnitude of the gradient is not changed, and we do not need to tune the learning rate as a consequence of introducing the uncertainty weight.

We propose a normality-guided algorithm with the uncertainty weight to improve policy for PPO and TRPO in Algorithm 1, but this method can be applied to various policy update algorithms as well.

## 5 Experiments

### 5.1 Experiment Setups

Our implementation is based on Spinning Up (Achiam, 2018), an open-source resource by OpenAI that provides implementations of several reinforcement learning algorithms and tools to help researchers get started with deep RL. For our experiments, we chose two representative deep reinforcement learning algorithms, Policy Optimization (PPO) (Schulman et al., 2017) and Trust Region Policy Optimization (TRPO) (Schulman et al., 2015a), to evaluate our method. The advantage is computed by GAE (Schulman et al., 2015b).

We use the default hyperparameters such as learning rate and batch size. All policies have a two-layer *tanh* network with 64 x 32 units, and all of the value function and the distributional value function has

Table 1: Ablation study of TRPO and PPO. Each entry represents an average return and a standard error, derived from 30 different settings, with each setting running for 100 episodes. $V^{D,\pi}$ (QR) represents the distributional value function trained with the Huber quantile regression. This table illustrates an improvement trend across the scores with the addition of each component from the baseline to MC-CLT. Also, MC-CLT results generally better than the Ensemble method, while using fewer weights and reduced training time. An asterisk (*) indicates a statistically significant improvement between MC-CLT and Ensemble method at a significance level of 0.05

|  | BipedalWalker | Hopper | HalfCheetah | Ant |
|---|---|---|---|---|
| TRPO | $115 \pm 3.2$ | $2724 \pm 31$ | $2749 \pm 37$ | $2811 \pm 16$ |
| TRPO $+V^{D,\pi}$ (QR) | $168 \pm 3.1$ | $2893 \pm 28$ | $2633 \pm 35$ | $2962 \pm 21$ |
| MC-CLT TRPO w/o $w$ | $186 \pm 3.1$ | $2835 \pm 27$ | $2984 \pm 39$ | $3107 \pm 18$ |
| Ensemble TRPO w/ $w$ | $180 \pm 2.7$ | $2742 \pm 29$ | $\mathbf{3212 \pm 44}$ | $\mathbf{3368 \pm 19}$ |
| MC-CLT TRPO (ours) | $\mathbf{195 \pm 2.6^*}$ | $\mathbf{2927 \pm 27^*}$ | $3172 \pm 43$ | $3329 \pm 20$ |
|  | Swimmer | Walker2d | InvertedDoublePendulum | LunarLander(C) |
| TRPO | $\mathbf{121 \pm 0.2}$ | $2558 \pm 26$ | $7969 \pm 109$ | $198 \pm 3.1$ |
| TRPO $+V^{D,\pi}$ (QR) | $117 \pm 0.2$ | $2398 \pm 25$ | $7878 \pm 112$ | $227 \pm 2.6$ |
| MC-CLT TRPO w/o $w$ | $118 \pm 0.4$ | $2616 \pm 30$ | $7662 \pm 114$ | $245 \pm 2.2$ |
| Ensemble TRPO w/ $w$ | $116 \pm 0.2$ | $2834 \pm 24$ | $\mathbf{8566 \pm 85}$ | $242 \pm 1.8$ |
| MC-CLT TRPO (ours) | $118 \pm 0.2^*$ | $\mathbf{2904 \pm 23^*}$ | $8404 \pm 93$ | $\mathbf{258 \pm 1.8^*}$ |
|  | BipedalWalker | Hopper | HalfCheetah | Ant |
| PPO | $179 \pm 2.9$ | $2973 \pm 27$ | $2902 \pm 42$ | $1595 \pm 10$ |
| PPO $+V^{D,\pi}$ (QR) | $208 \pm 2.7$ | $2778 \pm 30$ | $2574 \pm 33$ | $1742 \pm 11$ |
| MC-CLT PPO w/o $w$ | $215 \pm 2.4$ | $3184 \pm 23$ | $2870 \pm 39$ | $1743 \pm 13$ |
| Ensemble PPO w/ $w$ | $227 \pm 2.2$ | $3208 \pm 22$ | $3068 \pm 43$ | $1863 \pm 12$ |
| MC-CLT PPO (ours) | $\mathbf{236 \pm 2.3^*}$ | $\mathbf{3230 \pm 22}$ | $\mathbf{3110 \pm 39}$ | $\mathbf{1900 \pm 13^*}$ |
|  | Swimmer | Walker2d | InvertedDoublePendulum | LunarLander(C) |
| PPO | $\mathbf{122 \pm 0.1}$ | $2198 \pm 33$ | $8814 \pm 72$ | $229 \pm 2.1$ |
| PPO $+V^{D,\pi}$ (QR) | $112 \pm 0.8$ | $2037 \pm 34$ | $8913 \pm 64$ | $250 \pm 2.4$ |
| MC-CLT PPO w/o $w$ | $117 \pm 0.6$ | $2610 \pm 35$ | $9187 \pm 41$ | $268 \pm 1.7$ |
| Ensemble PPO w/ $w$ | $119 \pm 0.2$ | $2669 \pm 34$ | $8933 \pm 64$ | $256 \pm 1.8$ |
| MC-CLT PPO (ours) | $\mathbf{122 \pm 0.1^*}$ | $\mathbf{2716 \pm 33}$ | $\mathbf{9225 \pm 36^*}$ | $\mathbf{278 \pm 1.2^*}$ |

a two-layer *ReLU* network with 64 x 64 units or 128 x 128 units for all environments. All networks are updated with Adam optimizer (Kingma & Ba, 2014). We evaluate our method on continuous OpenAI gym Box2D (Brockman et al., 2016) and MuJoCo tasks (Todorov et al., 2012), as these environments have continuous action spaces and dense reward functions to use the normal approximation of MC-CLT (Theorem 1).

## 5.2 Detailed Ablation Analysis and Comparison

There exist three components that separate MC-CLT from the baseline. We perform an ablation study to analyze the impact of each. The first component involves substituting the standard value function with the distributional value $V^{D,\pi}$ trained with the Huber quantile regression. We refer to this as PPO $+V^{D,\pi}$ (QR) and TRPO $+V^{D,\pi}$ (QR). The second is the addition of the normal target for $V^{D,\pi}$, but without incorporating the uncertainty weight, denoted as (MC-CLT w/ $w$). Lastly, our method MC-CLT is utilizing the uncertainty weight in policy updates to prioritize more reliable samples (MC-CLT w/ $w$). Furthermore, we compare our method with the ensemble-based approach with uncertainty weight, as their approach also measures uncertainties and integrates them into policy updates in the same manner. Thus, we focuses on

Table 2: Training cost in terms of time and the required number of weights based on the baseline.

|  | Baseline | MC-CLT | Ensemble-based |
|---|---|---|---|
| Time | 1 | $1.5 - 1.9$ | $2.8 - 3.1$ |
| # weights | 1 | $2.0 - 2.2$ | 5 |

evaluating which type of uncertainty more accurately reflects sample reliability: the one derived from the network's accuracy or the one based on standard deviation.

Table 1 represents the experimental results across 8 environments. The first section of the table provides results from policies trained using TRPO, while the second section gives results from policies trained with PPO. As we incrementally add each component from the baseline to MC-CLT, a consistent trend of performance improvement is observed. Also, our method shows better results compared to the ensemble-based approach in general. In 10 out of the 16 settings, we achieve statistically significant improvements between MC-CLT and the Ensemble method, as indicated by an asterisk ($*$) in the table, at a significance level of 0.05. It's worth noting that the ensemble-based approach with TRPO exhibits better mean performances in three settings: HalfCheetah, Ant, and InvertedDoublePendulum. However, the differences are not statistically significant.

In addition, MC-CLT utilizes fewer weights and achieves faster training time, as shown in (Table 2). Table 2 presents a relative comparison with the other two methods based on the baseline. Specifically, when the training time for PPO is set to 1, the MC-CLT method requires 1.5 to 1.9 times more training time compared to the baseline. Similarly, the ensemble-based approach takes 2.8 to 3.1 times more time. Additionally, the MC-CLT method requires approximately 2.0 to 2.2 times more weights, while the ensemble-based approach uses 5 times as many parameters. We use 8 quantile bars for all experiments, as we empirically find that using 8 or more quantile bars provides smiliar satisfactory performance.

While MC-CLT requires hyperparameters for the number of quantile bars and the variance network, it necessitates fewer hyperparameters compared to previous methods (Lee et al., 2020; Mai et al., 2022). We observe satisfactory performance when the number of quantiles is set to $\geq 8$ and the size of the variance network equals to that of the distributional value function. In practice, we fix the number of quantiles at 8 for all experiments. Consequently, aside from the aspects controlling uncertainty, MC-CLT shares the same hyperparameters as the baseline method. The detailed hyperparameter settings are discussed in Appendix A.

### 5.3 Seeking Accuracy vs. Seeking Exploration

MC-CLT and the ensemble-based methods seek accuracy by reducing weights for samples with high uncertainty. This raises exploration concerns regarding states where the current distributional value function fails to predict accurately. If such states are not given sufficient weight, the policy may never cover them effectively. To address this concern, we conduct additional experiments where states with higher uncertainty receive higher weights to encourage the agent to explore these states more. We compute the weight with $w(s) = E * T + 0.5$, where $E = \sum_{i=0}^{N-1}(q_i(s) - q_i'(s))^2$, and run Algorithm 2 as in MC-CLT to find the temperature $T$ (where *left* and *right* are adjusted in the opposite direction from accuracy seeking). Table 3 shows the comparisons between accuracy-seeking and exploration-seeking approaches. In all experiments, the accuracy-seeking approach shows better performance. This is likely because the environments use highly engineered reward functions, providing sufficient feedback without requiring additional exploration. For tasks where discovering novel states is important, such as in Atari games (Bellemare et al., 2013), the exploration-seeking approach would have a positive effect on performance (Obando-Ceron et al., 2023; Schaul et al., 2022).

# 6 Conclusion

We have presented a distributional reinforcement learning (DRL) method in that output quantiles approximate the value distribution as a normal distribution under the mild assumption for Markov Chain Central Limit Theorem (Jones, 2004). Existing actor-critic algorithms that assess uncertainty typically employ multiple value functions (ensemble) to estimate variance. Given the presence of states with relatively high variance, we propose an alternative approach to measure uncertainty. Rather than relying on the variance, we evaluate how much the predicted quantiles are close to a normal.

Furthermore, when updating the distribution value function using the distributional Bellman optimality operator, we observe that the variance estimates tend to increase with each timestep. The discount factor of the operator yields a variance proportional to timestep. To address this, we guide the distributional value function by utilizing analytically computed quantile bars derived from returns and the variance network. Our method prioritizes accurate samples that are well-estimated by the current distributional value function, thereby promoting improved policy performance. Overall, our method exhibits better performance compared to the baselines in our evaluations. Additionally, it leverages a reduced number of weights, leading to faster training times than the ensemble-based method.

Although our proposed method has been primarily discussed in continuous tasks, where the assumption is that the return distribution follows a normal distribution, it also potentially works in other scenarios, such as tasks with discrete action spaces. Since we obtain the mean of the return through sampling, exactly as in original actor-critic methods, it remains unbiased, and the uncertainty is calculated solely based on the deviation from the target that our distributional value function is intended to fit. Additional insights and details are provided in Appendix E.

### Reproducibility Statement

We provide the hyperparameters used in our evaluations and all source code is available at `https://github.com/shashacks/MC_CLT`.

### Acknowledgement

The authors would like to thank the Ohio Supercomputer Center (Center, 1987) for providing the computational resources used in this research.

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

# A  Hyperparameters

Deep reinforcement learning algorithms pose a significant challenge for evaluation due to the inherent instability and stochasticity during training. To ensure a fair comparison with other algorithms, we attempted to maintain a consistent set of hyperparameters as much as possible. We follow Spinning Up (Achiam, 2018) default settings for learning rate, optimizer, etc. All policies have a two-layer *tanh* network with 64 x 32 units, and all of the value function and the distributional value function has a two-layer *ReLU* network with 64 x 64 units or 128 x 128 units for all environments.

Although Lee et al. (2020) suggest the formula to compute the uncertainty weight $w(s) = \sigma(-E * T) + w_{\min}$, it seems that finding the reasonable value for $T$ is not intuitive for various tasks. (Note that $E$ is computed as the normality error of the distributional value function in our method and $E$ is replaced with the standard deviation of the value functions in Lee et al. (2020)). We instead set a target uncertainty weight and perform a parametric search by adjusting $T$ to achieve that target weight (Appendix B). This process substitutes $T$ with the target weight $w_{\text{tar}}$ as a hyperparameter. $w_{\text{tar}}$ is chosen from $w_{\text{tar}} \in \{0.85, 0.9\}$, and $w_{\min}$ is chosen from $w_{\min} \in \{0.4, 0.5, 0.6\}$.

MC-CLT has the number of quantile output nodes of the distributional value function as a hyperparameter. We examine how performance is affected by the number of quantile bars, with values such as $N = 4, 8, 12$, and 20. We observe satisfactory performance when the number of quantile bars is set to $N \geq 8$, so we fix the number of quantiles at 8 for all experiments. Also, the ensemble-based method introduces the ensemble size as a hyperparameter. As Lee et al. (2020) and Mai et al. (2022) demonstrate that ensemble size $= 5$ shows the sufficient results, we set the ensemble size to 5 for these models.

In order to ensure a fair comparison with the ensemble methods, we selected $3-5$ best configurations, each of which is trained with ten different seeds. The numbers reported in the tables correspond to the results from the last 10 runs.

# B  Algorithms In Detail

Instead of manually tuning $T$ for each task, we dynamically adjust $T$ such that the average uncertainty lies within the range $[w_{\text{tar}} - \epsilon, w_{\text{tar}} + \epsilon]$. The following algorithm illustrates the parametric search employed to find such a $T$. While MC-CLT utilizes the error set $E$, the ensemble-based methods leverage the standard deviation set. The term $2(1 - w_{\min})$ ensures that each uncertainty weight lies within the range $(w_{\min}, 1.0)$.

---

**Algorithm 2** Parametric search to find the temperature $T$

---

1: **Input:** target weight $w_{\text{tar}}$, minimum weight $w_{\min}$, error set $E$
2: Initialize $left = 0$, $right = 2^{12}$
3: **while** $left \leq right$ **do**
4:     $T = (left + right)/2$
5:     $W = 2(1 - w_{\min})\sigma(-E * T) + w_{\min}$
6:     Compute the mean of uncertainty weight $w_{\text{avg}} = \text{Avg}(W)$
7:     **if** $w_{\text{avg}}$ is in the range $[w_{\text{tar}} - \epsilon, w_{\text{tar}} + \epsilon]$ **then**
8:         Use $T$ to compute the uncertainty weight
9:         **break**
10:    **else if** $w_{\text{avg}} \geq w_{\text{tar}} + \epsilon$ **then**
11:        $left = T$
12:    **else**
13:        $right = T$
14:    **end if**
15: **end while**

---

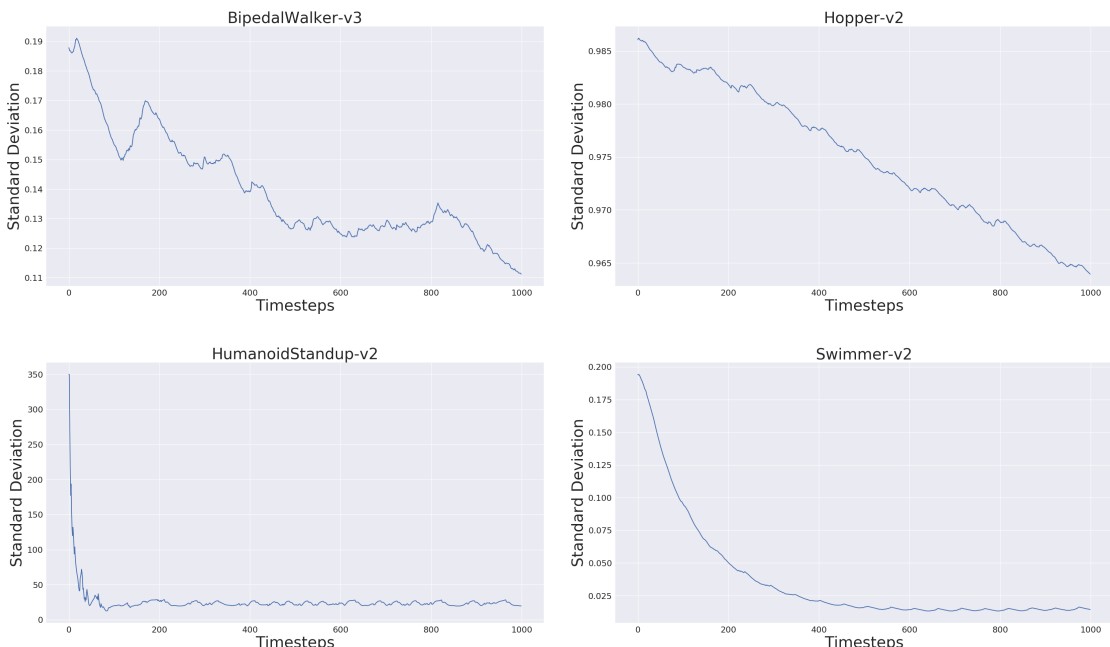

Figure 4: Changes in standard deviation over timestep. We train ensemble value functions and monitor the changes in standard deviations. The graphs indicate that the standard deviation consistently decreases as the timestep progresses.

## C  Variance Network

In practice, the variance network (Kendall & Gal, 2017) is trained to predict log variance $\ln\sigma_\psi^2$ instead of predicting variance directly. This avoids a division by zero or a negative value in the log term in Equation (6). However, after predicting log variance, it is necessary to retrieve the actual variance value through the exponential operation, i.e., $\sigma_\psi^2 = \exp(\ln\sigma_\psi^2)$.

However, in cases of environments having the large observation space, such as Ant-v2 or HumanoidStandup-v2, we frequently observe overflow issues during the exp operation, particularly in the early training epochs. Thus, we directly predict the variance by limiting the minimum value $\sigma_\psi^2 = \max(\epsilon, \sigma_\psi^2)$. We set $\epsilon = 0.0001$, and this resolves the overflow issue for the large observation space environments. We also compared the outcomes of the two methods on tasks with a smaller observation space. We couldn't see any meaningful difference between the two methods, so we conducted experiments for all environments by directly predicting the variance.

## D  Standard Deviation Change

Certain states display relatively higher variances compared to others. For instance, in fragile control tasks like Walker2d-v2, the states close to the *terminal* state typically exhibit higher variances. This is because once the agent overcomes the *terminal* states, several more successful actions are likely to be generated, increasing the sum of rewards (return). Additionally, the closer the state is to the initial timestep, the higher the variance. This correlation is apparent when comparing the changes in return from the initial and intermediate timesteps.

The graphs (Figure 4) have been obtained by applying exponential smoothing to the standard deviation of the trained value functions (the ensemble-based method), Notably, the standard deviation is observed to decrease over the timestep.

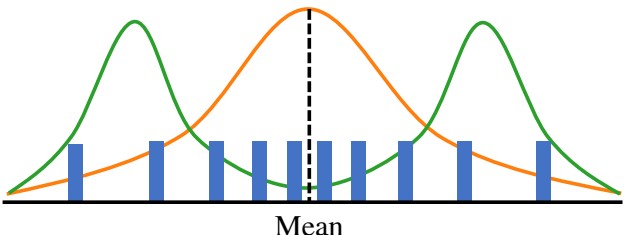

Mean

Figure 5: The green line depicts a non-normal complex distribution. The blue bars represent normal quantile bars, which are analytically computed using the mean of the complex distribution, with a variance obtained from a variance network. The orange line illustrates the line version of the normal distribution derived from the quantile bars.

Motivated by these observations, we propose a novel approach. Rather than decreasing the uncertainty weight of states with higher variance, we suggest a method for measuring the accuracy of the value function prediction. We then use this measure to obtain the uncertainty weight.

## E  Normality Discussion

This section addresses scenarios where tasks deviate from the normality condition. Our method does not compromise the policy update, even when the normality condition is not satisfied. Since we first compute the mean of the target quantiles from sampled episodes and then analytically compute the quantile bars (the variance is derived from the variance network), the mean represented by these quantile bars is unbiased.

In detail, suppose a complex green line distribution $\mathcal{D}$ is the true return distribution of a state $s$ under a policy $\pi$, as shown in Figure 5. A standard value function $V^\pi$, which predicts a single value (return) $V^\pi(s)$ for a state, estimates the mean of this complex distribution $\mathcal{D}$, indicated by the dotted line, through sampled episodes.

Rather than just outputting the single value, our distributional value function $V^{D,\pi}$ is trained to output blue quantile bars for $s$ under the same policy $\pi$, representing a normal distribution (the orange line). Although $V^{D,\pi}(s)$ represents the orange normal distribution, distinct from the green one, the mean of these quantile bars is also located at the dotted line position because the sampling mechanism are both the same. In other words, $\frac{1}{N}\sum_{i=0}^{N-1} q_i(s) = V^\pi(s)$, where $V^{D,\pi}(s) = \{q_0(s), q_1(s), ..., q_{N-1}(s)\}$. Given that this mean is used to update the policy, this alignment confirms that our approach *without uncertainty weights* is unbiased.

The question arises: *What happens when we use the uncertainty weights to update the policy in tasks deviating from the normality condition?* In this case, we argue that our method becomes a special case of Exploration by Random Network Distillation (RND) (Burda et al., 2018), in which a random neural network is provided, and another neural network is fitted to this random network. While the model is fitted, an exploration bonus is computed based on the differences between the two models. Differing from RND's approach of providing a random neural network, we provide a normal quantile target, expecting that $V^D$ will also produce quantile bars indicative of a normal distribution. We then measure how closely $V^D$'s predictions mirror the shape of a normal distribution.

Both approaches rely on the premise that the difference diminishes with sufficient learning. The distinction between RND and our method is that RND utilizes uncertainty as a bonus added to the reward, seeking novelty, whereas our approach is a risk-averse strategy by reducing the impact of gradients from high-uncertainty samples. These different search strategies can be chosen depending on the problem. Given that the reward function in control tasks, such as MuJoCo, is highly engineered, being risk-averse is more effective than seeking novelty, as the feedback from the reward function is sufficient, unlike sparse reward settings. We actually conduct experiments to check whether encouraging the agent to visit high-uncertainty states is beneficial for our environments (Table 3), and find that seeking exploration does not outperform in environments with highly engineered rewards.

Table 3: Seeking Accuracy vs. Seeking Exploration Table: We compute the uncertainty weight (our method) based on Equation 9. However, this raises concerns regarding exploration. To address this, we conduct an experiment using Equation (2) to compute the exploration weight, ensuring that states with high uncertainty are visited more frequently. Each entry represents an average return and a standard error, derived from 30 different settings, with each setting running for 100 episodes.

|  | BipedalWalker | Hopper | HalfCheetah | Ant |
|---|---|---|---|---|
| Exploration TRPO | $34 \pm 3.3$ | $2328 \pm 34$ | $2459 \pm 32$ | $1663 \pm 16$ |
| MC-CLT TRPO (ours) | $\mathbf{195 \pm 2.6}$ | $\mathbf{2927 \pm 27}$ | $\mathbf{3172 \pm 43}$ | $\mathbf{3329 \pm 20}$ |
|  | Swimmer | Walker2d | InvertedDoublePendulum | LunarLander(C) |
| Exploration TRPO | $36 \pm 2.2$ | $1086 \pm 33$ | $7628 \pm 112$ | $179 \pm 4.4$ |
| MC-CLT TRPO (ours) | $118 \pm 0.2$ | $\mathbf{2904 \pm 23}$ | $\mathbf{8404 \pm 93}$ | $\mathbf{258 \pm 1.8}$ |
|  | BipedalWalker | Hopper | HalfCheetah | Ant |
| Exploration PPO | $70 \pm 3.6$ | $2727 \pm 28$ | $2756 \pm 37$ | $1315 \pm 10$ |
| MC-CLT PPO (ours) | $\mathbf{236 \pm 2.3}$ | $\mathbf{3230 \pm 22}$ | $\mathbf{3110 \pm 39}$ | $\mathbf{1900 \pm 13}$ |
|  | Swimmer | Walker2d | InvertedDoublePendulum | LunarLander(C) |
| Exploration PPO | $44 \pm 0.8$ | $916 \pm 24$ | $9027 \pm 54$ | $230 \pm 2.8$ |
| MC-CLT PPO (ours) | $\mathbf{122 \pm 0.1}$ | $\mathbf{2716 \pm 33}$ | $\mathbf{9225 \pm 36}$ | $\mathbf{278 \pm 1.2}$ |

## F  Limitation

Our current method is primarily effective in stochastic continuous tasks with dense rewards based on the principle of MC-CLT. However, in environments where a value distribution is non-normal, alternative methods may be more effective for accurately capturing these non-normal value distributions. It's important to note that any limitations in our current approach don't lead to significant failures, as detailed in Appendix E.

## G  Potential Research Directions

The use of our method does not exclude the possibility of utilizing the ensemble technique. In fact, by combining our method with the ensemble, it becomes possible to leverage multiple distributional value functions. However, simply averaging the uncertainties from each function has shown no performance improvement compared to our method alone. Besides weighting policy updates, exploring alternative uses of uncertainties from each distributional value function would be an intriguing avenue for future research. For instance, in methods like Soft Actor-Critic (SAC) (Haarnoja et al., 2018) or Twin Delayed DDPG (TD3) (Fujimoto et al., 2018), where the minimum value is selected among two Q functions, the Q function with lower uncertainty could be chosen to utilize a more accurate value because our uncertainty weight represents the accuracy of the network.

