# OpenReview forum: "Normality-Guided Distributional Reinforcement Learning for Continuous Control"
_TMLR — Accepted by TMLR_

### Review · Reviewer_CHAg · 2024-06-27

**Summary Of Contributions:**

This paper focuses on distributional reinforcement learning, and proposes a method that assumes normality of the distribution of returns in continuous environments. Leveraging the assumption, the method is shown to outperform vanilla and distributional actor-critic algorithms on several continuous environments.

**Audience:**

No

**Broader Impact Concerns:**

I don't predict special ethical implications of the work.

**Claims And Evidence:**

No

**Requested Changes:**

I think the paper needs to improve significantly in terms of clarity and motivation. At the moment, neither the setting nor the objective of the paper seem clear to me, and it is somehow hard to follow and evaluate the work.

**Strengths And Weaknesses:**

The proposed method appears to outperform the baselines in several tasks in terms of final performance, while simultaneously using significantly less weights. The authors also claim that their method achieves this high performance with much less computations, but I could not find those results on the main paper. As far as I am aware, the method and the motivation are novel.

There are some important points that are not clear to me at this point:
- The paper start by showing that the distribution of returns closely relates to a normal, empirically, and recalls a theorem that as far as I understand is supporting that hypothesis. Does this finding contradict the currently used methods for distributional reinforcement learning? The arguments in Section 4.2 appear to hint that way, but this was not clear in the paper.
- In light of the above, the goal of the paper is not clear. If the goal is solely to provide a method that outperforms the currents methods for distributional reinforcement learning, it should be stated earlier in the paper.

As the biggest weakness of the paper, I would point that it is not very clear. For instance, in Figure 3, we don't have any scale or values on the x and y axis, so it is very hard to understand and validate the plot. Figures 1 and 2 are also difficult to interpret, despite the extensive description. I believe the notation may be a bit dense and that the paper would benefit from also describing the results in a more abstract way. I also think the setting is confusing: the MDP is introduced as infinite-horizon, but there is reference to "done" states, which are not described. Additionally, if the MDP is infinite-horizon, I don't understand very well what is meant by later states, or states in later time steps, and remaining time-steps.

Minor:
- Figure 2 appears on page 5, while only being referred in page 7, whereas Figure 3 appears on page 6 while being referred on page 5 (so I believe the figures should be switched, and appear on top of the page they are referred to).
- There is a reference to a p-value of 0.05, when I believe it is meant a significance level of 0.05.
- I also don't think it is clear for a general reader what is Spinning Up (page 8).

---

> ### Author Response · Authors · 2024-10-27
>
> Thank you for your valuable review. We answer your questions and have revised the draft as follows:
>
> **Question 1**
> > The paper start by showing that the distribution of returns closely relates to a normal, empirically, and recalls a theorem that as far as I understand is supporting that hypothesis. Does this finding contradict the currently used methods for distributional reinforcement learning? The arguments in Section 4.2 appear to hint that way, but this was not clear in the paper.
>
> => Previous distributional RL methods produce variances in the value distribution that are not consistent with the Markov chain central limit theorem (MC-CLT). While MC-CLT indicates that variance decreases over timesteps, previous works show increasing variance because they follow the distributional Bellman optimality operator. Additionally, we leverage the normality property from MC-CLT to compute uncertainty weights. We can compute the locations of quantile bars for normal distributions, which is not present in previous works.
>
> **Qeustion 2**
> > The proposed method appears to outperform the baselines in several tasks in terms of final performance, while simultaneously using significantly less weights. The authors also claim that their method achieves this high performance with much less computations, but I could not find those results on the main paper.
>
> => Section 5.2 (3rd paragraph) discusses the training cost. Table 2 compares the training time of our method to the baseline and the ensemble-based method. Our method takes 1.5 to 1.9 times longer to train, while the ensemble-based method takes 2.8 to 3.1 times longer than the baseline. Also, our method uses fewer weights than the ensemble-based approach; ours uses 2.0 to 2.2 times the baseline weights, while the ensemble-based method uses 5 times.
>
> **Weakness 1**
> > In light of the above, the goal of the paper is not clear. If the goal is solely to provide a method that outperforms the currents methods for distributional reinforcement learning, it should be stated earlier in the paper.
>
> => We have strengthened the introduction section to improve clarity. 1) The current distributional Bellman optimality operator increases the variance of the distribution as the timestep progresses, as introduced in Section 4.2. This contradicts Theorem 3.3 (MC-CLT), which states that the variance should decrease over time. We propose an alternative target (normal quantiles) for the distributional value function, guided by the variance predicted by the variance network. 2) As we make our distributional value function estimate quantile bars representing normal distributions, we can measure how closely it approximates a normal distribution and use this as uncertainty.
>
> **Minor 1**
> > In Figure 3, we don't have any scale or values on the x and y axis, so it is very hard to understand and validate the plot.
>
> => We have separated the graph into 4 figures, and each plot has x and y scales.
>
> **Minor 2**
> > I believe the notation may be a bit dense and that the paper would benefit from also describing the results in a more abstract way. I also think the setting is confusing: the MDP is introduced as infinite-horizon, but there is reference to "done" states, which are not described. Additionally, if the MDP is infinite-horizon, I don't understand very well what is meant by later states, or states in later time steps, and remaining time-steps.
>
> => We would like to clarify that terminal states are compatible with infinite-horizon MDPs, as detailed in [1] (Chapter 3.4). We formulate our problem as an infinite-horizon MDP to develop a method that generalizes to both episodic and non-episodic tasks without requiring prior knowledge of the horizon, consistent with standard definitions in the literature [1, 2, 3].
>
> To clarify terminology, we have changed *done* states to *terminal* states. *Terminal* states refer to states where the agent can no longer continue the task (e.g., when it falls). Additionally, environments like MuJoCo and Box2D have predefined maximum timesteps, such as the 1000-timestep limit used in our experiments, which also results in a terminal state. Here, *latter timesteps* refer to timesteps that occur after timestep $t$, such as $t+1, t+2,$ and so on.
>
> **Minor 3**
> > Figure 2 and Figure 4 positions, p-value 0.05, and Spinning Up
>
> => We have switched the figure positions, changed *p-value 0.05* to *significance level of 0.05,* and added more information about Spinning Up.
>
> References
>
> [1] Reinforcement Learning: An Introduction, Sutton and Barto, 2018\
> [2] Trust Region Policy Optimization, Schulman et al., 2015\
> [3] High-Dimensional Continuous Control Using Generalized Advantage Estimation, Schulman et al., 2015

---

### Review · Reviewer_Vrpn · 2024-09-15

**Summary Of Contributions:**

The authors built on the successes of distributional reinforcement learning, and emprically observed that the distribution for the value distrbution is close to a normal distribution. Based on this observation, they proposed to separately estimate the variance for each state, and subsequently designed a weighted policy update strategy. The experiments demonstrated the superior performance of the proposed algorithm.

**Audience:**

Yes

**Claims And Evidence:**

Yes

**Requested Changes:**

1. Please address the first comment regarding the weakness above. Perhaps you can run more experiment on other MuJoCo domains?
2. When measuring uncertainty, why did you choose N=4 bars? How does this number affect the overall performance?
3. The authors mentioned "correct locations" just above Equation (9). To my understanding, they are still estimated from samples so why are they "correct"?

**Strengths And Weaknesses:**

Strengths
1. The idea itself looks interesting and reasonable. The authors first identified the simlarity to a normal distribution and then were able to measure the uncertainty, which was used to derive weighted based updates.
2. The authors conducted empiricall studies on a few MuJoCo domains, and the experiments demonstrated the improvement compared with other methods.
3. This paper is presented well and mostly easy to follow.

Weaknesses
1. One claim the authors claimed is that the policy updates need to focus on the states the current value function predicts correctly. I can see the reasoning behind it. On the other hand, I have some concerns regarding the states the current value fuction fails to predict: If such states are not given enough weights, the policy would never cover them well. Subsequently, exploration in this algorithm will be an issue.
2. Some ablation studies are necessary to validate a few claims.

---

> ### Author Response · Authors · 2024-10-27
>
> Thank you for your valuable comments. We answer your questions and have revised the draft as follows:
>
> **Weakness 1**
> > One claim the authors claimed is that the policy updates need to focus on the states the current value function predicts correctly. I can see the reasoning behind it. On the other hand, I have some concerns regarding the states the current value fuction fails to predict: If such states are not given enough weights, the policy would never cover them well. Subsequently, exploration in this algorithm will be an issue.
>
> => We have added Section 5.4 to address this concern. We compare accuracy-seeking (our method) with exploration-seeking methods. In exploration-seeking, weights are computed to encourage the agent  to visits states with high uncertainty, in contrast to our method. Table 3 shows the results, and accuracy-seeking demonstrates better performance than exploration-seeking across all environments.
>
> **Weakness 2**
> > Some ablation studies are necessary to validate a few claims.
>
> => Starting with basic baselines like PPO, we have covered all ablation cases. DPPO represents the distributional value function from PPO. MC-CLT without the uncertainty weight refers to DPPO with a guided target, but without using the uncertainty weight to update the policy. When the uncertainty weight is used to update the policy, it becomes MC-CLT PPO.
>
> **Question 1**
> > When measuring uncertainty, why did you choose $N=4$ bars? How does this number affect the overall performance?
>
> => We used $N=4$ to explain how the quantile target is concretely computed with Figure 3 and specific numbers in Section 4.4. In practice, we used $N=8$ because we observed satisfactory performance when the number of quantiles is set to $\geq 8$. We have strengthened this explanation in the experiment section and Appendix A.
>
>
> **Question 2**
> > The authors mentioned "correct locations" just above Equation (9). To my understanding, they are still estimated from samples so why are they "correct"?
>
> => As you pointed out, we agree that using the word "correct" is excessive, so we have removed this part.

---

### Review · Reviewer_5uVb · 2024-10-15

**Summary Of Contributions:**

This paper observes that the value distribution in continuous tasks is close to normal under certain conditions and use this property to design a normality-guided method for updating the distributional value function, which provides quantile targets conforming to a normal distribution to solve the variance problem caused by the traditional distributional Bellman optimality operator. Experiments are conducted using PPO and TRPO algorithms on multiple continuous control tasks and compared with standard algorithms and an ensemble-based method.

**Audience:**

Yes

**Claims And Evidence:**

Yes

**Requested Changes:**

NA

**Strengths And Weaknesses:**

Strengths:
1. The observation of the normality distribution for continuous tasks and the corresponding design provides a new perspective for distributional reinforcement learning.
2. The idea of the proposed approach is compatible with many DRL structures and can be applied to different policy update algorithms.
3. The proposed approach uses fewer weights and has a shorter training time, reducing computational costs and resource consumption. And it outperforms the ensemble-based method in most tested environments, significantly improving policy performance with statistically significant improvements in 10/16 settings.

Weaknesses:
There is no clear weakness.

---

> ### Author Response · Authors · 2024-10-27
>
> Thank you for your positive feedback and for recognizing the strengths of our work. We have refined the manuscript by incorporating insights and addressing the concerns raised by other reviewers.

---

### Decision · Action_Editor_cceX · 2025-05-08

**Recommendation:** Accept as is

**Comment:**

As mentioned above, after addressing reviewer feedback, the paper clearly presents a simple, yet novel, approach to distributional RL that the RL community can leverage for further advancements. The majority of reviewers support acceptance, with which I agree.

**Audience:**

This paper would be of interest to many in the RL community.

**Claims And Evidence:**

This paper empirically demonstrates that the value distribution learned by distributional RL algorithms in continuous control tasks often ends up close to a normal distribution. Given this, the authors propose a novel distributional RL method that exploits this finding, and empirically demonstrate that it yields performance improvements.

After addressing reviewer feedback, the paper is clearer, well-scoped, and provide adequate empirical evidence for their claims.